# 🌿 MINT-1T:
# Scaling Open-Source Multimodal Data by 10x: A Multimodal Dataset with One Trillion Tokens

**Anas Awadalla**[1,2*]   **Le Xue**[2]   **Oscar Lo**[1]   **Manli Shu**[2]
**Hannah Lee**[1]   **Etash Guha**[1]   **Matt Jordan**[4]   **Sheng Shen**[5]   **Mohamed Awadalla**[1]
**Silvio Savarese**[◇,2,3]   **Caiming Xiong**[◇,2]   **Ran Xu**[◇,2]   **Yejin Choi**[◇,1]   **Ludwig Schmidt**[◇,1]

[1] University of Washington, [2] Salesforce Research, [3] Stanford University,
[4] University of Texas at Austin, [5] University of California, Berkeley, [◇]Senior Authors

## Abstract

Multimodal interleaved datasets featuring free-form interleaved sequences of images and text are crucial for training frontier large multimodal models (LMMs). Despite the rapid progression of open-source LMMs, there remains a pronounced scarcity of large-scale, open-source multimodal interleaved datasets. In response, we introduce 🌿 MINT-1T, the most extensive and diverse open-source **M**ultimodal **INT**erleaved dataset to date. 🌿 MINT-1T comprises of one trillion text tokens and 3.4 billion images, a 10x scale-up from existing open-source datasets. Additionally, we include previously untapped sources such as PDFs and ArXiv papers. As scaling multimodal interleaved datasets requires substantial engineering effort, sharing the data curation process and releasing the dataset greatly benefits the community. Our experiments show that LMMs trained on MINT-1T rival the performance of models trained on the previous leading dataset, OBELICS. We release our data at `https://github.com/mlfoundations/MINT-1T`.

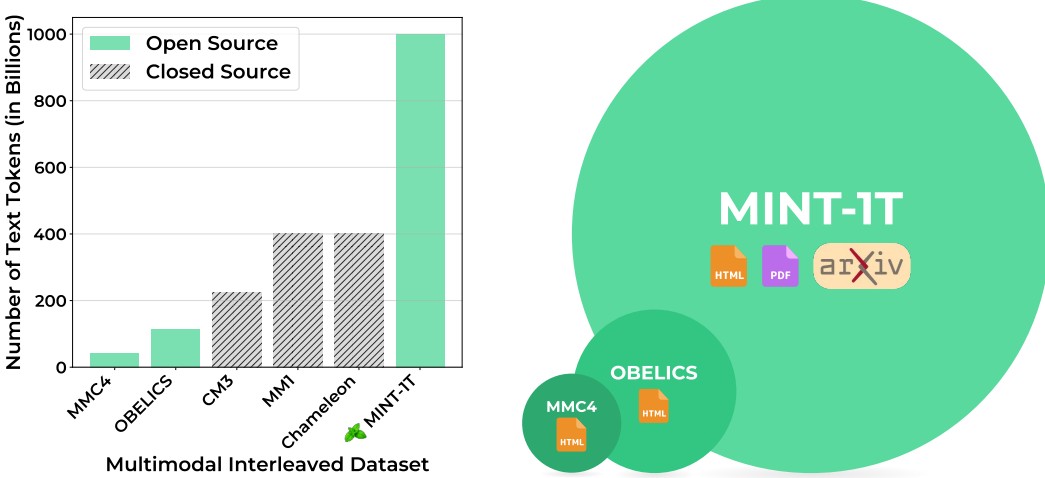

Figure 1: 🌿 MINT-1T is a one trillion token multimodal interleaved pre-training dataset. This is the largest dataset of its kind and is more diverse than previous open-source datasets.

---

*Work done while interning at Salesforce Research

38th Conference on Neural Information Processing Systems (NeurIPS 2024) Track on Datasets and Benchmarks.

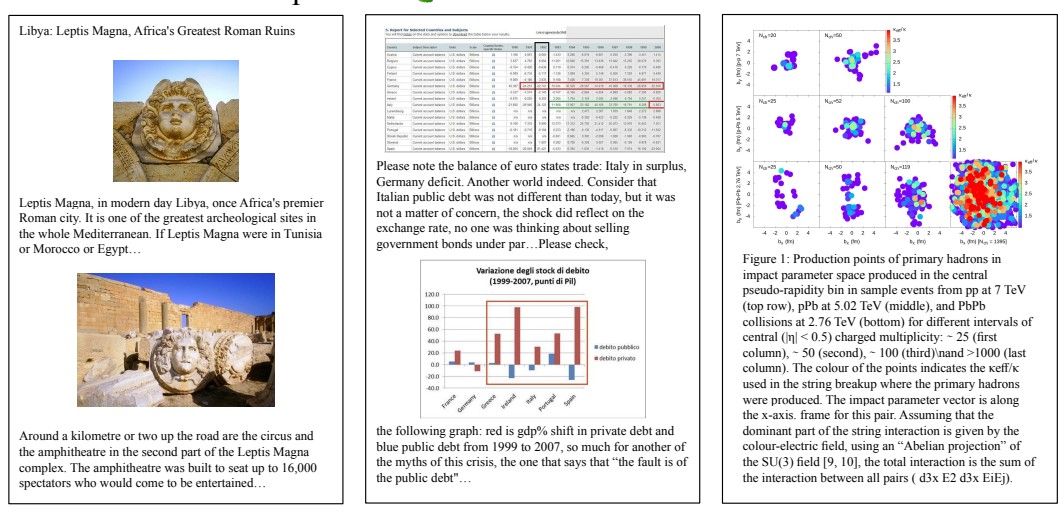

Examples of 🌿MINT Multimodal Documents

Figure 2: Samples of multimodal document from the HTML (Left), PDF (Middle), and ArXiv (Right) subsets of 🌿 MINT-1T with each document containing a sequence of images interleaved with text. Previous work has shown that interleaved data improves question-answering performance in the context of Flamingo-style models [Laurençon et al., 2023] and for training large multimodal models with a strong performance on both text-only and multimodal benchmarks [McKinzie et al., 2024]. MINT-1T is the first open-source work to scale interleaved datasets to one trillion text tokens and collect interleaved documents from PDFs and ArXiv at large scales. Samples in this figure are text truncated due to space.

## 1 Introduction

Large open-source pre-training datasets have been important artifacts for the research community in studying data engineering and training transparent, open-source models. In the text domain, we have seen how early works such as C4 [Raffel et al., 2019] and The Pile [Gao et al., 2020] were integral for the community to train the first set of open-source large language models (GPT-J [Wang and Komatsuzaki, 2021], GPT-Neo [Black et al., 2021], and others). These works also set the stage for subsequent works that improved on data filtering methods and scale. Similar trends hold in the image-text space large-scale open-source datasets led to innovation on better data curation methods such as Data filtering networks [Fang et al., 2023], T-MARS [Maini et al., 2023], and others.

We are seeing a major shift from frontier labs to train large multimodal models (LMMs) [Google, 2023, Meta, 2024, Achiam et al., 2023] which require large multimodal interleaved datasets—comprising of free-form sequences of images and texts (an example of interleaved documents can be found in Figure 2). As the capabilities of frontier models advance rapidly, there is an increasing gap in the multimodal training data between closed- and open-source models. Existing open-source multimodal interleaved datasets are smaller and less diverse compared to their text-only counterparts and are sourced only from HTML documents, limiting the breadth and variety of data. This restriction hampers the development of robust open-source LMMs and creates a disparity between the capabilities of open and closed-source models.

To bridge this gap, we built 🌿 MINT-1T (**M**ultimodal **INT**erleaved), the largest and most diverse open-source multimodal interleaved dataset to date. MINT-1T contains a total of one trillion text tokens and three billion images, which are sourced from diverse sources like HTML/PDFs/ArXiv. Before MINT-1T, the largest open-source dataset in this area was OBELICS [Laurençon et al., 2023], a 115 billion text token dataset with 353M images sourced only from HTML.

Our contributions with 🌿 MINT-1T are as follows:

**Data Engineering** Scaling this multimodal interleaved data presents more of an engineering challenge than building either text-only or image-text pair datasets. We are handling much larger document sizes, and the original ordering of images and text must be preserved.

| Dataset | # Tokens | # Images | # Docs | Open Source | Data Sources |
|---|---|---|---|---|---|
| CM3 | 223B | 373M | 10.7M | ✗ | HTML |
| Multimodal-C4 | 43B | 571M | 101M | ✓ | HTML |
| OBELICS | 115B | 353M | 141M | ✓ | HTML |
| MM1 | 400B | – | – | ✗ | HTML |
| Chameleon | 400B | – | – | ✗ | HTML |
| 🌿 MINT-1T | 1.02T | 3.42B | 1054M | ✓ | HTML/PDFs/ArXiv |

Table 1: **Survey of multimodal interleaved datasets:** Current open-source interleaved datasets are smaller than proprietary datasets. Beyond HTML data sources, 🌿 MINT-1T uniquely includes data from PDFs and ArXiv documents.

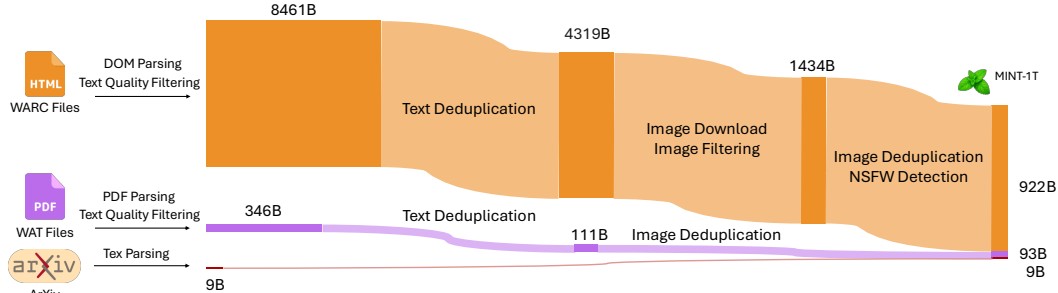

Figure 3: The filtering process for 🌿 MINT-1T shows how tokens are removed throughout the data pipeline for HTML, PDFs, and ArXiv papers.

**Diversity** We are the first work in the multimodal interleaved space to gather high-quality multimodal documents at large scales from sources like CommonCrawl PDFs and ArXiv.

**Model Experiments** Our experiments demonstrate that LMMs trained on 🌿 MINT-1T not only match but potentially surpass the performance of models trained on the best existing open-source dataset, OBELICS, while offering a tenfold increase in scale.

## 2 Dataset Construction

MINT-1T curates a large-scale open-source dataset that taps into more diverse sources of interleaved documents such as PDFs and ArXiv papers. This section outlines our methods for sourcing multi-modal documents, filtering low-quality documents, deduplicating data, and removing not safe for work and undesirable content. The final dataset contains **922 billion (B)** HTML tokens, **93B** PDF tokens, and **9B** ArXiv tokens (for more detailed numbers and filtering ratios refer to Figure 3).

### 2.1 Sourcing Large Quantities of Multimodal Documents

#### 2.1.1 HTML Pipeline

We follow OBELICS's method for extracting interleaved multimodal documents from CommonCrawl WARC files by parsing each WARC entry's DOM tree. While OBELICS only processed documents from February 2020 to February 2023 CommonCrawl dumps, we have expanded our document pool to include HTML documents from May 2017 to April 2024 (note that we use the full dumps from October 2018 to April 2024 and partial dumps from earlier years). Following OBELICS, we filter out any documents that contain no images or more than thirty images or any images with URLs that include inappropriate substrings such as *logo, avatar, porn, and xxx*.

#### 2.1.2 PDF Pipeline

We source PDF documents from CommonCrawl WAT files from February 2023 to April 2024 dumps. Initially, we extract all the PDF links from these dumps. We then attempt to download and read PDFs

using PyMuPDF [2]. We discard PDF documents that are more than 50MB large (as they likely contain predominantly large images) and PDFs that are over 50 pages long. We exclude pages that contain no text and extract a reading order for the remaining pages. Reading order is obtained by finding the bounding box of all text blocks on a page, clustering the blocks based on columns, and ordering the blocks from top left to bottom right. Images are anchored in the sequence based on the proximity between the image's bounding box and text blocks on the same page. We discuss the limitations of this approach in Appendix A.1.

### 2.1.3 ArXiv pipeline

ArXiv interleaved documents are built from the LaTeX source code. We use TexSoup [3] to find figure tags and interleave the images with the paper text. For multi-file papers (i.e. where each section is written in a different Tex file), we identify the main Tex file and replace input tags with the contents of its file. We additionally clean up the the LaTex code removing imports, bibliography, tables, and citation tags. As ArXiv is already a highly curated data source, we do not perform any of the filtering and deduplication described in the rest of this section.

## 2.2 Text Quality Filtering

In line with practices established by RefinedWeb [Penedo et al., 2023], Dolma [Soldaini et al., 2024], and FineWeb [Penedo et al., 2024], we avoid using model-based heuristics for text filtering. This approach has proven to scale efficiently for text-only models. Initially, we eliminate non-English documents using Fasttext's [Joulin et al., 2017] language identification model (with a confidence threshold of 0.65). Additionally, documents with URLs containing NSFW substrings were removed to exclude pornographic and undesirable content. We apply text filtering methods from RefinedWeb, specifically removing documents with excessive duplicate n-grams or those identified as low quality in using MassiveText [Rae et al., 2021] rules.

## 2.3 Image Filtering

After obtaining the curated set of PDFs and HTML files, we attempt to download all image URLs in the HTML dataset, discarding any non-retrievable links and removing documents that have no valid image links. We remove images smaller than 150 pixels to avoid noisy images such as logos and icons and images larger than $20,000$ pixels as those usually correspond to off-topic images. For HTML documents, we remove images with an aspect ratio greater than two to remove low-quality images such as advertisement banners. However, for PDFs, we adjust this threshold to three to preserve scientific figures and tables, which are often erroneously excluded by stricter criteria.

## 2.4 Safety Filtering

**NSFW Image Filtering**    We apply an NSFW image detector [Laborde] to all images in our dataset. If we find that a document contains a single NSFW image, we discard the entire document.

**Personally Identifiable Information Removal**    To mitigate the risk of personal data leakage, we anonymize email addresses and IP addresses in our text data. Following FineWeb, we replace emails with templates such as "email@example.com" and IPs with randomly generated non-functional IPs.

## 2.5 Deduplication

To remove duplicated content, we perform paragraph and document text deduplication within each CommonCrawl snapshot and image deduplication to remove repetitive, uninformative images such as icons and logos. All deduplication steps are done separately for each data source.

---

[2] `https://github.com/pymupdf/PyMuPDF-Utilities/blob/master/text-extraction/multi_column.py`

[3] `https://github.com/alvinwan/TexSoup`

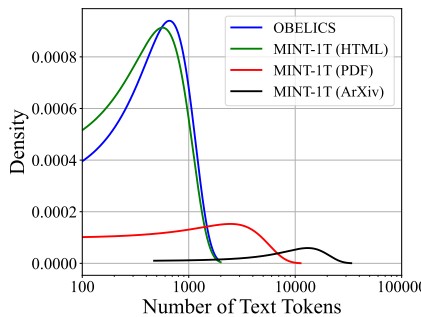

Figure 4: We plot the distribution of text tokens per document for OBELICS and MINT-1T. We observe that the HTML subset of MINT-1T follows a similar distribution to OBELICS, but the PDFs and ArXiv have significantly longer lengths.

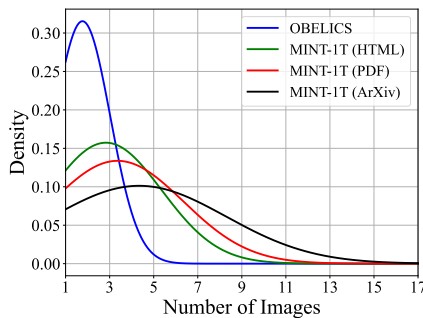

Figure 5: Documents in MINT-1T, on average, contain more images than OBELICS. Our HTML subset contains more images than OBELICS, and we see that within MINT-1T, PDFs are slightly more image dense than HTML, with ArXiv being the most image dense.

### 2.5.1 Paragraph and Document Deduplication

Following Dolma's methodology [Groeneveld, 2023], we use a Bloom Filter for efficient text deduplication. We set the false positive rate to $0.01$ for the bloom filter and deduplicate 13-gram paragraphs (indicated through double newline delimiters) from each document. If more than $80\%$ of a document's paragraphs are duplicates, we discard the entire document.

### 2.5.2 Removing Common Boilerplate Text

Post-paragraph deduplication, we notice that short common boilerplate sentences in HTML documents, such as "Skip to content" or "Blog Archive," remain. To remove these noisy sentences, we run exact paragraph deduplication on $2\%$ of each CommonCrawl snapshot, in line with CCNet [Wenzek et al., 2019]; doing this at small scales ensures we mostly remove just common boilerplate text.

### 2.5.3 Image Deduplication

Within each CommonCrawl snapshot, we remove frequently occurring images based on SHA256 hashes. Rather than strict deduplication, we follow Multimodal-C4 [Zhu et al., 2023] by only removing images that appear more than ten times within a snapshot. Consistent with OBELICS [Laurençon et al., 2023], we remove repeated images within a single document and keep only the first occurrence.

### 2.6 Infrastructure

Throughout the data processing, we had access to an average of 2,350 CPU cores from a mixture of 190-processor and 90-processor nodes. In total, we used ~4.2M CPU hours to build this dataset.

## 3 Analysis

### 3.1 Comparing Document Composition in MINT-1T with OBELICS

In assessing the composition of interleaved datasets, two key characteristics are examined: the distribution of text tokens per document and the number of images per document. For this analysis, we randomly sampled 50,000 documents from both OBELICS and each data source in MINT-1T. We use GPT-2's tokenizer to calculate the number of text tokens. We remove outliers, excluding documents that fall outside the 1.5 * interquartile range for the number of text tokens and images. As shown in Figure 4, the HTML subset of MINT-1T aligns closely with the token distribution seen in OBELICS. However, documents sourced from PDFs and ArXiv tend to be longer than HTML documents on overage, highlighting the benefits of sourcing data from diverse sources. Figure 5

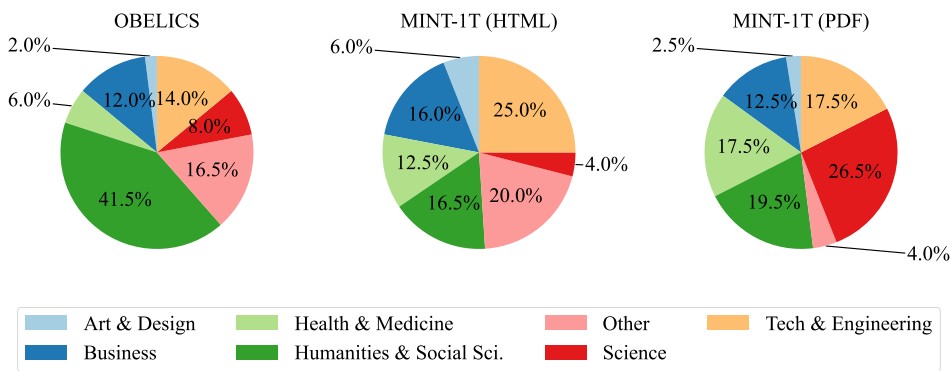

Figure 6: **Document domain distribution:** The percentage of documents from each domain in MMMU for OBELICS and subsets of MINT-1T. We find two interesting trends: (1) The majority of documents in OBELICS are related to the *Humanities and Social Sciences*; this trend isn't found in MINT-1T's HTML subset. (2) The majority of PDF documents are *Science* related.

examines the image density across all documents, revealing that PDFs and ArXiv documents contain more images compared to HTML documents, with ArXiv samples being the most image dense.

## 3.2 How Do Different Data Sources Improve Document Diversity?

An important motivation for expanding the pool of multimodal documents beyond HTML is the improvement of domain coverage. To quantify the diversity and depth of this coverage, we employ a Latent Dirichlet Allocation [Campbell et al., 2003] (LDA) model trained on 100,000 documents sampled from the OBELICS dataset, the HTML subset of MINT-1T, and the PDF subset (excluding ArXiv) from MINT-1T to get 200 topics. We then use GPT-4 to classify the set of words to identify the dominant domains – such as Health & Medicine, Science, Business, Humanities, History, etc. – based on MMMU domains.

Our analysis reveals distinct trends in domain distribution:

**OBELICS:** This dataset shows a pronounced concentration in "Humanities and Social Sciences". This may be attributed to its data construction process, which involves filtering out documents that do not resemble Wikipedia articles, thus potentially altering the distribution to more general knowledge and humanities-focused content.

**MINT-1T's HTML Subset:** In contrast to OBELICS, the HTML subset of MINT-1T is not strongly biased towards any specific domain, suggesting a broader and more balanced domain representation.

**MINT-1T's PDF Subset:** There is a higher proportion of "Science and Technology" documents within the PDF documents of MINT-1T. This trend is likely due to the nature of scientific communication, where PDFs are the preferred format for sharing detailed research papers and technical reports.

## 4 Model Experiments

### 4.1 Training Setup

In this section, we outline the architecture of the LMMs we train, the hyper-parameters used, and the methods to evaluate the multimodal interleaved abilities of the models.

**Modeling** Our architecture is adopted from XGen-MM [Xue et al., 2024]. We use the ViT-H vision encoder with resolution 378 from Data Filtering Networks [Fang et al., 2023] and pass the patch embeddings into a six layer perceiver resampler [Alayrac et al., 2022]. Each image is represented as 128 tokens. The pooled patch embeddings are interleaved with the text token embeddings and passed into Phi-3 mini language model [Abdin et al., 2024]. We keep the vision encoder frozen while training the resampler and the language model. We use a batch size of, on average, 1.8M multimodal tokens. For all of our training runs, we use 2,000 warmup steps with a maximum learning rate of $5 * 10^{-5}$ and cosine learning rate decay. We also apply 0.05 weight decay to all trainable parameters.

All of our training is done using the OpenFlamingo [Awadalla et al., 2023] codebase. We train all of our models on 32 H100 GPUs for a total of 1,920 GPU hours per experiment.

**Training Data** For all experiments, we train our model on $50\%$ image-text captioning batches and $50\%$ multimodal interleaved batches. We sample a maximum of $2048$ multimodal tokens from each interleaved document and 340 tokens from each image-text sample. As in Flamingo Alayrac et al. [2022], we add an `<|endofchunk|>` token to indicate the end of an adjacent image-text sequence. During training, we randomly drop 50% of single-image interleaved documents to upsample multi-image documents. For our image-text dataset, we use a mixture of internal curated caption datasets.

## 4.2 Evaluation Setup

We assess a model's capability to reason about multimodal interleaved sequences through its in-context learning abilities and multi-image reasoning performance.

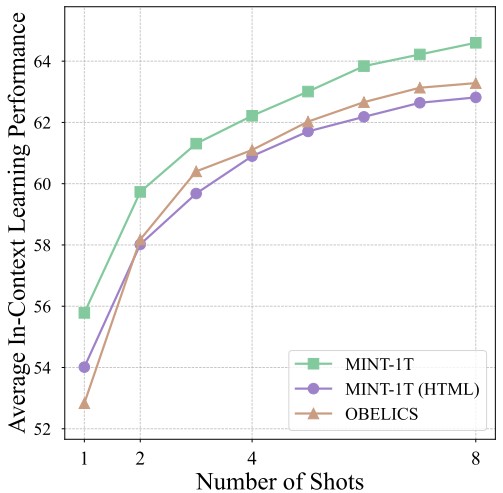 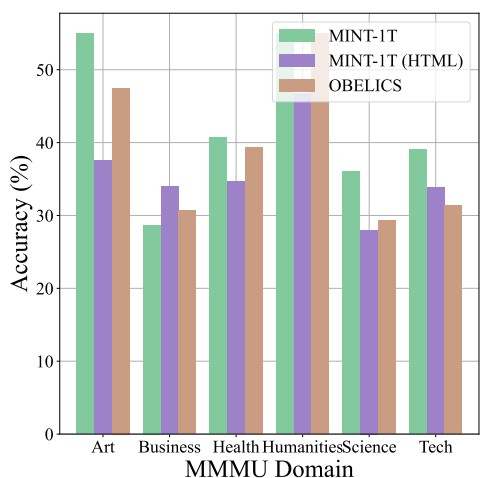

(a) Average performance on captioning and visual question answering benchmarks when using one to eight demonstrations. The model trained on MINT-1T performs better across all demonstrations in comparison to the HTML portion of MINT-1T or OBELICS.

(b) MINT-1T outperforms OBELICS on many domains. We improve on OBELICS in *Science* and *Technology* domains as the PDF subset of MINT-1T contains a large representation of these domains. MINT-1T also performs better than OBELICS on *Art*.

**In-context Learning** The models are evaluated on four-shot and eight-shot in-context learning performance on various captioning benchmarks (COCO (Karpathy test) [Lin et al., 2014] and TextCaps (validation) [Sidorov et al., 2020]) and visual question answering datasets (VQAv2 (validation) [Agrawal et al., 2015], OK-VQA (validation) [Marino et al., 2019], TextVQA (validation) [Singh et al., 2019], and VizWiz (validation) [Gurari et al., 2018]). For all evaluations, we randomly sample demonstrations from the training set. Our reported scores are averaged over multiple evaluation runs where we randomize demonstrations. We find that performance is sensitive to the chosen prompts, so we ablate through different prompts for each task and choose the prompt that performs best. The list of prompts we use and generation parameters can be found in Appendix A.2.

**Multi-image Reasoning** We additionally evaluate models on MMMU [Yue et al., 2024](containing both single and multi-image questions) and Mantis-Eval [Jiang et al., 2024](all multi-image questions) to probe a model's multi-image reasoning abilities beyond in-context learning evaluations.

## 4.3 Experiments

### Training on HTML Documents

We first evaluate how the HTML portion of MINT-1T compares to OBELICS, as OBELICS is the previous leading interleaved dataset and is also curated from HTML documents. We train two models on the HTML portions of MINT-1T and OBELICS for 10B multimodal tokens total and assess their

| Model | Shots | Datasets | | | | | | Average |
|---|---|---|---|---|---|---|---|---|
| | | COCO | TextCaps | OKVQA | TextVQA | VQAv2 | VizWiz | |
| OBELICS | 4 | **108.1** ± 0.91 | 80.4 ± 0.96 | 48.4 ± 0.79 | 42.4 ± 0.74 | 61.8 ± 0.12 | 26.0 ± 0.28 | 61.2 ± 0.28 |
| | 8 | 109.4 ± 0.71 | 83.9 ± 0.36 | 49.6 ± 0.08 | 43.8 ± 0.43 | 62.3 ± 0.42 | 30.9 ± 1.10 | 63.3 ± 0.24 |
| MINT-1T (HTML) | 4 | 105.4 ± 0.99 | 79.2 ± 1.17 | 48.0 ± 0.12 | 43.7 ± 0.35 | 63.5 ± 0.11 | 25.6 ± 0.44 | 60.9 ± 0.27 |
| | 8 | 107.9 ± 0.58 | 81.9 ± 1.29 | 49.7 ± 0.37 | 44.2 ± 0.30 | 64.3 ± 0.15 | 30.3 ± 0.87 | 63.0 ± 0.28 |
| 🌿 MINT-1T | 4 | 107.0 ± 0.13 | 79.7 ± 0.62 | **49.7** ± 0.19 | **45.0** ± 0.39 | **64.9** ± 0.07 | **27.5** ± 0.32 | **62.3** ± 0.13 |
| | 8 | 108.8 ± 0.34 | **84.3** ± 0.51 | **51.1** ± 0.18 | **45.6** ± 0.10 | **66.1** ± 0.09 | 31.8 ± 0.53 | **64.6** ± 0.14 |

Table 2: **In-context learning evaluations:** We train three 4.6B multimodal models on 10B tokens of a mixture of image-text and interleaved samples documents. Models are evaluated using four and eight in-context learning examples and each evaluation is run for three trials. We report the mean performance and standard deviation. Statistically significant performance gaps are bolded and equally performing models are underlined.

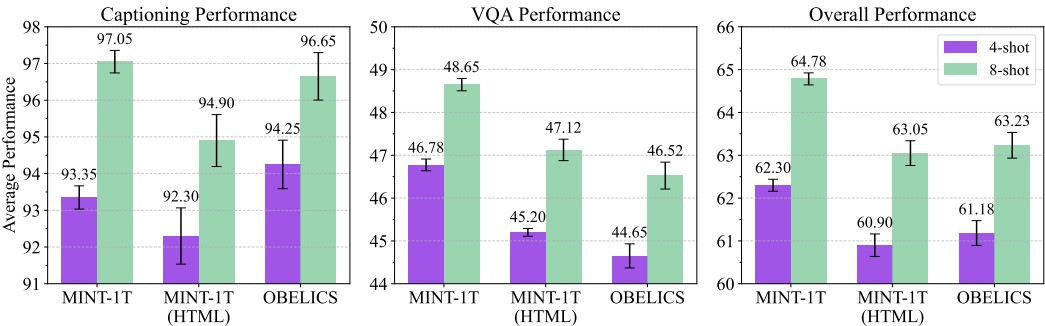

Figure 8: Performance on captioning and visual question answering (VQA) tasks. The HTML subset of MINT-1T outperforms OBELICS on VQA tasks but performs worse on captioning benchmarks.

in-context learning performance. In Table 2, we present 4-shot and 8-shot performance on common benchmarks; the model trained on MINT-1T HTML documents performs better than OBELICS on VQA tasks but worse on captioning benchmarks. On average OBELICS performs slightly better than MINT-1T (HTML). We explore how model architecture impacts this result in Section 4.5.

**Adding PDF and ArXiv documents** Subsequently, we train on MINT-1T's full data sources, with a mixture of HTML, PDF, and ArXiv documents. We sample 50% of our interleaved documents from HTML, 45% from PDFs, and 5% from ArXiv. We train for a total of 10B multimodal tokens. As seen in Table 2, the model trained on the full MINT-1T data mixture outperforms OBELICS and MINT-1T (HTML) on most in-context learning benchmarks. On more complex multimodal reasoning benchmarks, the MINT-1T model outperforms OBELICS on MMMU but performs worse on Mantis-Eval.

| Model | Datasets | |
|---|---|---|
| | MMMU | Mantis-Eval |
| OBELICS | 37.6 | **44.2** |
| MINT-1T (HTML) | 35.2 | 41.5 |
| 🌿 MINT-1T | **41.3** | 43.3 |

Table 3: Performance on image reasoning benchmarks MMMU and Mantis-Eval.

## 4.4 Fine-grained Trends

**How Does In-Context Learning Performance Scale with Demonstrations?** We evaluate models' in-context learning performance when prompted with one to eight demonstrations. We run a single trial per shot count for each evaluation benchmark described in Section 4.2. As seen in Figure 7a, we find that the model trained on MINT-1T outperforms the model trained on the HTML subset of MINT-1T and OBELICS on all shots. Moreover, we find that the MINT-1T (HTML) model performs slightly worse than OBELICS.

**Performance on Captioning and Visual Question Answering Tasks** In Figure 8, we present the average in-context learning performance on captioning and visual question answering (VQA) benchmarks. OBELICS outperforms all MINT-1T variants on four shot captioning benchmarks and performs slightly worse to MINT-1T on eight shot captioning. However, we find that on VQA

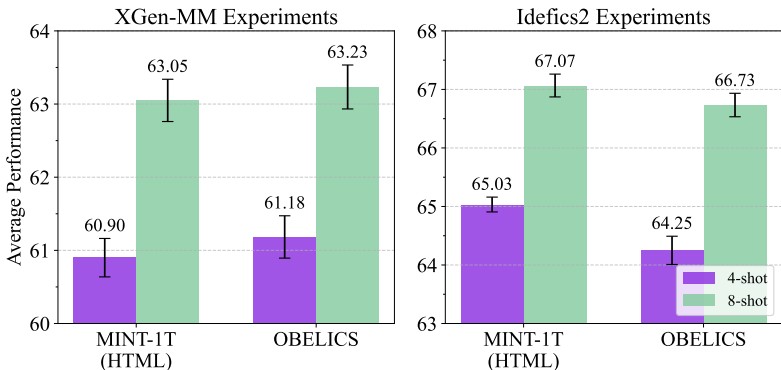

Figure 9: **Impact of architecture:** On in-context learning benchmarks, XGen-MM models perform marginally better when trained on OBELICS compared to MINT-1T's HTML subset. In contrast, Idefics2 models show a slight advantage for MINT-1T (HTML) over OBELICS.

| Model | Shots | Datasets | | | | | | Average |
| | | COCO | TextCaps | OKVQA | TextVQA | VQAv2 | VizWiz | |
|---|---|---|---|---|---|---|---|---|
| OBELICS | 4 | 110.2 ± 0.38 | 83.1 ± 1.28 | 52.8 ± 0.04 | 46.7 ± 0.23 | 63.8 ± 0.10 | 28.9 ± 0.50 | 64.3 ± 0.24 |
| | 8 | 111.8 ± 1.04 | 86.6 ± 0.31 | **54.8** ± 0.04 | 46.9 ± 0.02 | 64.1 ± 0.09 | 36.2 ± 0.51 | 66.7 ± 0.20 |
| MINT-1T (HTML) | 4 | **110.9** ± 0.01 | **84.8** ± 0.05 | 52.9 ± 0.26 | **47.0** ± 0.31 | **65.6** ± 0.04 | 29.0 ± 0.64 | **65.0** ± 0.12 |
| | 8 | 111.3 ± 0.05 | **87.5** ± 0.11 | 54.1 ± 0.49 | **48.1** ± 0.12 | 64.8 ± 1.04 | 36.6 ± 0.13 | **66.9** ± 0.19 |

Table 4: **Idefics2 model results:** We compare OBELICS and MINT-1T (HTML), when training an Idefics2 LMM. Models are evaluated using four and eight in-context learning examples, with each evaluation run for two trials. We report the mean performance and standard deviation.

benchmarks MINT-1T is significantly better than both baselines. We also see that MINT-1T (HTML) also outperforms OBELICS on VQA tasks.

**Performance on Different Domains** A motivation for including diverse domains in MINT-1T is to improve model generalization. In Figure 7b, we break down performance on MMMU for each domain. With the exception of the Business domain, MINT-1T outperforms OBELICS and MINT-1T (HTML). We highlight the performance increase on Science and Technology domains for MINT-1T and speculate that this can be attributed to the prevalence of these domains in ArXiv and PDF documents.

## 4.5  Impact of Model Architecture

Our experiments, presented in Section 4.3, use XGen-MM's architecture. Unlike with large language models, the design space for multimodal models is much more diverse with many architectures for aligning a vision encoder to a language model. Naturally, we were curious if our results would hold for other popular training setups.

To investigate this, we replicate our training experiments using Idefics2's architecture. Idefics2 differs from XGen-MM in that it freezes a non-instruction finetuned large language model and adds LoRA [Laurenccon et al., 2024] matrices on all linear layers. For our Idefics2 reproduction we use the Mistral-7B-v0.3 language model and DFN ViT-H vision encoder with resolution 384. Unlike Idefics2, we do not experiment with flexible image resolution in training and keep the vision encoder completely frozen. We present in-context learning results for Idefics2 model experiments in Table 4. We find that MINT-1T's HTML subset performs better than OBELICS with notable gains on TextVQA, VQAv2, and TextCaps. We highlight the performance gap difference between XGen-MM and Idefics2 ablations in Figure 9. One key difference in the Idefics2 experiments is that the HTML subset of MINT-1T performs much more competitively on captioning benchmarks in comparison to OBELICS.

# 5 Related Work

## 5.1 Multimodal Interleaved Data

Large-scale multimodal interleaved datasets were first presented in Flamingo [Alayrac et al., 2022] and CM3 [Aghajanyan et al., 2022]. Kosmos [Huang et al., 2023] showed similar properties and was followed by Multimodal-C4 [Zhu et al., 2023] and OBELICS [Laurençon et al., 2023], the first open-source multimodal interleaved datasets. More recent work such as Chameleon [Meta, 2024] and MM1 [McKinzie et al., 2024] have scaled OBELICS to train state-of-the-art multimodal models. A complementary line of work, Mantis [Jiang et al., 2024] and MIMIC-IT [Li et al., 2023] builds interleaved instruction tuning datasets. Similarly, Multimodal Arxiv [Li et al., 2024a] builds high quality captioning and instruction tuning data from ArXiv papers.

## 5.2 Large Open-source Pre-training Datasets

Large, high-quality pre-training datasets are the backbone of open-source research. In image-text datasets, where preliminary works [Schuhmann et al., 2021, Byeon et al., 2022, Schuhmann et al., 2022, Gadre et al., 2023] focused on scaling image-text datasets to billions of samples and has been crucial for training strong open-source multimodal models [Ilharco et al., 2021, Sun et al., 2023a]. Similarly in language modeling, large datasets like Pile [Gao et al., 2020], Redpajama [Computer, 2023], RefinedWeb [Penedo et al., 2023], Dolma [Soldaini et al., 2024], Datacomp-LM [Li et al., 2024b], and FineWeb [Penedo et al., 2024] have been crucial for training fully transparent open-source models.

## 5.3 Large Multimodal Models

The past year has seen a large influx in strong large multimodal models (LMMs). There is a line of work that seeks to pre-train existing language models on large-scale multimodal interleaved and image-text datasets. This was first presented Flamingo [Alayrac et al., 2022] and adopted by open-source models such as OpenFlamingo [Awadalla et al., 2023], Idefics [Laurençon et al., 2023], and Emu [Sun et al., 2023b]. More recent works like Idefics2 [Laurenccon et al., 2024], MM1 [McKinzie et al., 2024], VILA [Lin et al., 2024], and XGen-MM [Xue et al., 2024] have also adopted similar data mixtures. A separate line of work aligns large language models to vision encoders using high-quality instruction-tuning data and image-text datasets such as LLaVA [Liu et al., 2023a,b], InstructBLIP [Dai et al., 2023], QwenVL [Bai et al., 2023], Yi-VL [Young et al., 2024], MiniCPM-V [Hu et al., 2024], and more. Moreover, the latest generation of large models such as GPT4-o [Achiam et al., 2023], Gemini [Google, 2023], and Chameleon [Meta, 2024] are trained on multimodal data from the start.

# 6 Limitations and conclusion

In this work, we present MINT-1T, the first open-source trillion token multimodal interleaved dataset and an important component for training large multimodal models. We believe MINT-1T will be a valuable resource for the research community to do open science on multimodal interleaved datasets. An important consideration when releasing large datasets is to avoid exposing harmful content from the web. We were careful about filtering out personally identifiable information and not safe for work content from MINT-1T. However, future work should explore more ways to improve the safety of multimodal internet data. Moreover, subsequent work should train models on larger subsets of MINT-1T, build filtering methods to improve data quality, and curate multimodal sequences from other sources.

## Acknowledgments and Disclosure of Funding

We thank Srinath Meadusani and Lavanya Karanam for working on infrastructure, Jeffrey Li and Alex Fang for helpful discussions regarding data filtering and deduplication, Irena Gao for leading the development of the new OpenFlamingo training codebase which we use, Honglu Zhou for maintaining the evaluation code, James Park and Marianna Nezhurina for helpful discussions regarding PDF parsing, and Paul Josel for helping us with figure design.

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

# A  Appendix

## A.1  PDF Parsing Issues

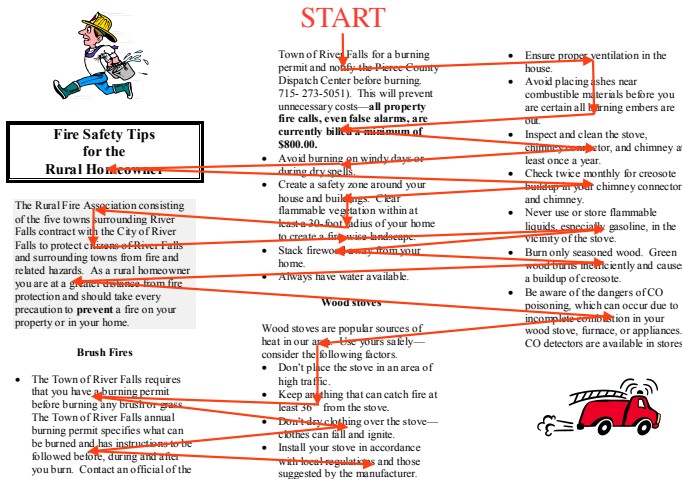

Figure 10: Results of incorrect PDF reading order extraction. As shown in this figure, a common issue is incorrectly reading across column boundaries. Other issues include incorrectly starting the reading from the rightmost column.

We noticed two limitations with using the PyMuPDF package for parsing PDFs: 1) we sometimes are unable to detect images 2) as PDFs have arbitrary layouts, in some cases we extract an incorrect reading order (see Figure 10). This compromise between speed and accuracy is a noted limitation, and future work should focus on developing fast, robust methods for determining the reading order across diverse document layouts.

## A.2  Evaluation Details

In this section we describe in detail how we prompt models for each evaluation tasks and the generation parameters we apply.

**In-Context Learning**  We ablated multiple prompts for captioning and visual question answering benchmarks and found prompts that worked best across all Phi-3 mini based large multimodal models. For COCO captioning we used the prompt `"A short description of this image in one sentence:"`. As TextCaps is a more OCR intensive captioning task we used `"A short description of this image that describes visual content and explicitly spells out any text it contains:"`. For VQAv2, VizWiz, and OKVQA we used the prompt `"Question: <question> Short answer in one phrase or single word:"` and for TextVQA we used `"Question: <question> Short answer in a very single phrase:"`. For all tasks we used greedy decoding with a maximum generation length of 56 tokens. For each evaluation run we experiment by separating demonstrations using `<|endofchunk|>` or double newline delimiters. We empirically find that MINT trained models perform better when using double newline delimiters while OBELICS trained model performs better using `<|endofchunk|>` separators. We do not report the best score for each evaluation trial by ablating over separators but rather choose the overall best prompt for a model based on aggregate scores and report results using that prompt for all evaluations.

**MMMU**  Our MMMU evaluation in Section 4 is based on the implementation from VLMEvalKit Contributors [2023][4]. We modify the codebase to support our model definition. We use the default prompting strategy from this codebase, which appends `"Answer:   "` to the question. Our results on MMMU are obtained in a zero-shot way on the validation split. We use greedy sampling for language generation with the maximum output token length set to 32.

**Mantis-Eval**  We use the official Mantis-Eval Jiang et al. [2024] codebase[5] to evaluate our model on this benchmark. Mantis-Eval consists of two categories of questions: "multi-choice" and "short-answer" questions. At inference time, we prepend each question with a one-shot demonstration. The demonstrations are provided in the eval codebase by the authors of Mantis. For each question type, we use a fixed example as the demonstration. The one-shot demonstration and the actual question are separated with the `<|endofchunk|>` token. For language generation, we use the configuration provided by Mantis-Eval, which samples model output with beam search with num_beams=3, and the maximum output token length is 512.

### A.3  Datasheet

#### A.3.1  Motivation

- **For what purpose was the dataset created?** Was there a specific task in mind? Was there a specific gap that needed to be filled? Please provide a description.

  MINT-1T is built for pre-training large multimodal models that can process multiple interleaved images and text. We fill a gap in the open-source space where there is a lack of large scale multimodal interleaved pre-training datasets.

- **Who created the dataset (e.g., which team, research group) and on behalf of which entity (e.g., company, institution, organization)?**

  The dataset is created by a team from the University of Washington, Salesforce Research, Stanford University, University of Texas at Austin, and University of California Berkeley.

- **Who funded the creation of the dataset?** If there is an associated grant, please provide the name of the grantor and the grant name and number.

  Compute for building MINT-1T and training ablation models came from Salesforce Research.

- **Any other comments?**

#### A.3.2  Composition

- **What do the instances that comprise the dataset represent (e.g., documents, photos, people, countries)?** Are there multiple types of instances (e.g., movies, users, and ratings; people and interactions between them; nodes and edges)? Please provide a description.

  The released dataset contains HTML, PDF, and ArXiv multimodal documents.

- **How many instances are there in total (of each type, if appropriate)?**

  In total there are 1054.3 million (M) documents (1029.4M HTML documents, 24.0M PDF documents, and 0.87M ArXiv documents).

- **Does the dataset contain all possible instances or is it a sample (not necessarily random) of instances from a larger set?** If the dataset is a sample, then what is the larger set? Is the sample representative of the larger set (e.g., geographic coverage)? If so, please describe how this representativeness was validated/verified. If it is not representative of the larger set, please describe why not (e.g., to cover a more diverse range of instances, because instances were withheld or unavailable).

  MINT-1T's HTML documents are filtered from CommonCrawl WARC dumps from 2017 to 2024 based on text quality, the presence of duplicated and undesirable content, and availability of images for downloading. MINT-1T PDF documents are filtered from CommonCrawl WAT dumps from 2023 to 2024 also based on text quality and the presence of duplicated and undesirable content. MINT-1T ArXiv documents is a subset of all ArXiv documents.

---

[4]`https://github.com/open-compass/VLMEvalKit`
[5]`https://github.com/TIGER-AI-Lab/Mantis`

- **What data does each instance consist of?** "Raw" data (e.g., unprocessed text or images) or features? In either case, please provide a description.

  For HTML documents we release the document's text along with image urls. For PDFs and ArXiv we also release the text along with the url for the PDF and a list of image reference numbers used to parse the images from the documents.

- **Is there a label or target associated with each instance?** If so, please provide a description.

  Not applicable.

- **Is any information missing from individual instances?** If so, please provide a description, explaining why this information is missing (e.g., because it was unavailable). This does not include intentionally removed information, but might include, e.g., redacted text.

  None.

- **Are relationships between individual instances made explicit (e.g., users' movie ratings, social network links)?** If so, please describe how these relationships are made explicit.

  Not applicable.

- **Are there recommended data splits (e.g., training, development/validation, testing)?** If so, please provide a description of these splits, explaining the rationale behind them.

  There is only a training split for MINT-1T.

- **Are there any errors, sources of noise, or redundancies in the dataset?** If so, please provide a description.

  None.

- **Is the dataset self-contained, or does it link to or otherwise rely on external resources (e.g., websites, tweets, other datasets)?** If it links to or relies on external resources, a) are there guarantees that they will exist, and remain constant, over time; b) are there official archival versions of the complete dataset (i.e., including the external resources as they existed at the time the dataset was created); c) are there any restrictions (e.g., licenses, fees) associated with any of the external resources that might apply to a dataset consumer? Please provide descriptions of all external resources and any restrictions associated with them, as well as links or other access points, as appropriate.

  MINT-1T is not self-contained and relies on downloading external image urls to collect the full dataset. There are no guarantees these urls will remain available over time as link rot is a common problem for large scale image datasets. Moreover, as this dataset is over 300TB large, it is infeasible for us to host the full datasets (with images included). There are no restrictions regarding downloading images from these external urls.

- **Does the dataset contain data that might be considered confidential (e.g., data that is protected by legal privilege or by doctor-patient confidentiality, data that includes the content of individuals' non-public communications)?** If so, please provide a description.

  Yes, while MINT-1T is sourced from the public web, there might be content that is considered confidential.

- **Does the dataset contain data that, if viewed directly, might be offensive, insulting, threatening, or might otherwise cause anxiety?** If so, please describe why.

  Yes it is possible such content can be found in MINT-1T. We take many steps to remove such content by removing urls with substrings that might be associated with not safe for work and undesirable content. We mask all email addresses and IP addresses to avoid leaking such data. Furthermore we run an image classifier over our entire dataset removing pornographic and undesirable images.

If the dataset does not relate to people, you may skip the remaining questions in this section.

- **Does the dataset identify any subpopulations (e.g., by age, gender)?** If so, please describe how these subpopulations are identified and provide a description of their respective distributions within the dataset.

  We do not explicitly identify any subpopulations but it is possible this information can be extracted from the dataset.

- **Is it possible to identify individuals (i.e., one or more natural persons), either directly or indirectly (i.e., in combination with other data) from the dataset?** If so, please describe how.

  Yes if it is present on the public web then it is possible to find images of individuals as well as text about specific individuals. We masked personally identifiable information such as emails and IP addresses to remove personal data from MINT-1T.

- **Does the dataset contain data that might be considered sensitive in any way (e.g., data that reveals race or ethnic origins, sexual orientations, religious beliefs, political opinions or union memberships, or locations; financial or health data; biometric or genetic data; forms of government identification, such as social security numbers; criminal history)?** If so, please provide a description.

  Yes if it is present on the public web then it is possible such data is found in MINT-1T.

- **Any other comments?**

### A.3.3 Collection Process

- **How was the data associated with each instance acquired?** Was the data directly observable (e.g., raw text, movie ratings), reported by subjects (e.g., survey responses), or indirectly inferred/derived from other data (e.g., part-of-speech tags, model-based guesses for age or language)? If the data was reported by subjects or indirectly inferred/derived from other data, was the data validated/verified? If so, please describe how.

  The data was not sourced from human responses. The data comes from CommonCrawl dumps and is filtered using a series of rules-based heuristics and deduplication methods.

- **What mechanisms or procedures were used to collect the data (e.g., hardware apparatuses or sensors, manual human curation, software programs, software APIs)? How were these mechanisms or procedures validated?**

  The data is parsed from HTML and PDF documents that come from CommonCrawl and ArXiv dumps. We apply quality filtering and deduplication methods that were previously validated in other large scale datasets. We validate our methods by training multiple large multimodal models.

- **If the dataset is a sample from a larger set, what was the sampling strategy (e.g., deterministic, probabilistic with specific sampling probabilities)?**

  Not applicable.

- **Who was involved in the data collection process (e.g., students, crowdworkers, contractors) and how were they compensated (e.g., how much were crowdworkers paid)?**

  No crowdworkers or contractors were involved in the data collection processes; only the authors of this work were involved.

- **Over what timeframe was the data collected?** Does this timeframe match the creation timeframe of the data associated with the instances (e.g., recent crawl of old news articles)? If not, please describe the timeframe in which the data associated with the instances was created.

  The data was collected from January 2024 to June 2024. We include web data from 2013 to 2024.

- **Were any ethical review processes conducted (e.g., by an institutional review board)?** If so, please provide a description of these review processes, including the outcomes, as well as a link or other access point to any supporting documentation.

  No ethical review has been conducted.

If the dataset does not relate to people, you may skip the remaining questions in this section.

- **Did you collect the data from the individuals in question directly, or obtain it via third parties or other sources (e.g., websites)?**

  Data was obtained from a third party web crawl, CommonCrawl, for HTML and PDF documents and directly from ArXiv S3 buckets for ArXiv documents.

- **Were the individuals in question notified about the data collection?** If so, please describe (or show with screenshots or other information) how notice was provided, and provide a link or other access point to, or otherwise reproduce, the exact language of the notification itself.

  No the individuals were not explicitly notified.

- **Did the individuals in question consent to the collection and use of their data?** If so, please describe (or show with screenshots or other information) how consent was requested and provided, and provide a link or other access point to, or otherwise reproduce, the exact language to which the individuals consented.

  We build our dataset on top of CommonCrawl which respects robots.txt files and therefore individuals that don't want their data to be crawled would not be included in our dataset. We conduct additional steps to mask personally identifiable information and remove not safe for work images.

- **If consent was obtained, were the consenting individuals provided with a mechanism to revoke their consent in the future or for certain uses?** If so, please provide a description, as well as a link or other access point to the mechanism (if appropriate).

  Yes upon release of the dataset we will provide a Google form to remove samples from MINT-1T.

- **Has an analysis of the potential impact of the dataset and its use on data subjects (e.g., a data protection impact analysis) been conducted?** If so, please provide a description of this analysis, including the outcomes, as well as a link or other access point to any supporting documentation.

  No.

- **Any other comments?**

### A.3.4 Preprocessing/cleaning/labeling

- **Was any preprocessing/cleaning/labeling of the data done (e.g., discretization or bucketing, tokenization, part-of-speech tagging, SIFT feature extraction, removal of instances, processing of missing values)?** If so, please provide a description. If not, you may skip the remaining questions in this section.

  Yes, the dataset was preprocessed to remove low quality text, duplicate documents/text portions, and remove not safe for work and low quality images.

- **Was the "raw" data saved in addition to the preprocessed/cleaned/labeled data (e.g., to support unanticipated future uses)?** If so, please provide a link or other access point to the "raw" data.

  No, we do not provide access to the raw data as it may contain offensive content and is not useful for training capable multimodal models.

- **Is the software that was used to preprocess/clean/label the data available?** If so, please provide a link or other access point.

  NSFW image detection - `https://github.com/GantMan/nsfw_model`
  Language identification - `https://fasttext.cc/`
  Text quality filters - `https://github.com/huggingface/datatrove`
  Deduplication - `https://github.com/allenai/bff`
  PDF parsing - `https://github.com/pymupdf/PyMuPDF`
  HTML parsing - `https://github.com/huggingface/OBELICS`

- **Any other comments?**

### A.3.5 Uses

- **Has the dataset been used for any tasks already?** If so, please provide a description.

  Yes, we used MINT-1T to train large multimodal models to validate the data quality against competitors.

- **Is there a repository that links to any or all papers or systems that use the dataset?** If so, please provide a link or other access point.

  We intend to release such a repository once we make the dataset public.

- **What (other) tasks could the dataset be used for?**
  The dataset can be used for other training objectives like generating interleaved images and text sequences or for building multimodal document retrieval systems.

- **Is there anything about the composition of the dataset or the way it was collected and preprocessed/cleaned/labeled that might impact future uses?** For example, is there anything that a dataset consumer might need to know to avoid uses that could result in unfair treatment of individuals or groups (e.g., stereotyping, quality of service issues) or other risks or harms (e.g., legal risks, financial harms)? If so, please provide a description. Is there anything a dataset consumer could do to mitigate these risks or harms?
  We follow previous works in not using text quality classifier models as they have been shown to favor text from certain demographics. While we take steps to reduce biases and risks in our dataset (see A.3.2), we encourage dataset consumers to further filter the data based on their use case.

- **Are there tasks for which the dataset should not be used?** If so, please provide a description.
  MINT-1T was built to make research into large multimodal models more accessible. Using the dataset to train models that ingest or generate personally identifying information (such as images of people's faces and other sensitive content) as well as military applications are all inappropriate use cases of MINT-1T.

- **Any other comments?**

### A.3.6 Distribution

- **Will the dataset be distributed to third parties outside of the entity (e.g., company, institution, organization) on behalf of which the dataset was created?** If so, please provide a description.
  Yes the dataset will be released to the public through the Huggingface interface.

- **How will the dataset will be distributed (e.g., tarball on website, API, GitHub)?** Does the dataset have a digital object identifier (DOI)?
  The dataset will be distributed as JSON shards where each entry contains the document's text and links to images. We will provide a DOI when released.

- **When will the dataset be distributed?**
  We plan to release the dataset July 2024.

- **Will the dataset be distributed under a copyright or other intellectual property (IP) license, and/or under applicable terms of use (ToU)?** If so, please describe this license and/or ToU, and provide a link or other access point to, or otherwise reproduce, any relevant licensing terms or ToU, as well as any fees associated with these restrictions.
  We release MINT-1T under a CC-BY-4.0 licence.

- **Have any third parties imposed IP-based or other restrictions on the data associated with the instances?** If so, please describe these restrictions, and provide a link or other access point to, or otherwise reproduce, any relevant licensing terms, as well as any fees associated with these restrictions.
  No.

- **Do any export controls or other regulatory restrictions apply to the dataset or to individual instances?** If so, please describe these restrictions, and provide a link or other access point to, or otherwise reproduce, any supporting documentation.
  No.

- **Any other comments?**

### A.3.7 Maintenance

- **Who will be supporting/hosting/maintaining the dataset?**
  This work's authors will actively respond to issues in MINT-1T and maintain the dataset.

- **How can the owner/curator/manager of the dataset be contacted (e.g., email address)?**
  The owners can be contacted through email or the Issues tab in the Huggingface interface.

- **Is there an erratum?** If so, please provide a link or other access point.

  Yes, the erratum will be present on the README of the Huggingface dataset page.

- **Will the dataset be updated (e.g., to correct labeling errors, add new instances, delete instances)?** If so, please describe how often, by whom, and how updates will be communicated to dataset consumers (e.g., mailing list, GitHub)?

  Yes the dataset will be updated by the authors of this work. We will communicate updates through the README in the Huggingface interface.

- **If the dataset relates to people, are there applicable limits on the retention of the data associated with the instances (e.g., were the individuals in question told that their data would be retained for a fixed period of time and then deleted)?** If so, please describe these limits and explain how they will be enforced.

  While the individuals were not told about data retention, we will provide a form for opting data out of MINT-1T.

- **Will older versions of the dataset continue to be supported/hosted/maintained?** If so, please describe how. If not, please describe how its obsolescence will be communicated to dataset consumers.

  Older versions of the dataset can be found using the Git history of the dataset repository. We will communicate the issues with previous versions in the erratum.

- **If others want to extend/augment/build on/contribute to the dataset, is there a mechanism for them to do so?** If so, please provide a description. Will these contributions be validated/verified? If so, please describe how. If not, why not? Is there a process for communicating/distributing these contributions to dataset consumers? If so, please provide a description.

  We will accept contributions to the dataset through the Pull Request mechanism of the Huggingface interface.

- **Any other comments?**

## A.4   License and Author Statement

We release MINT-1T under a CC-BY-4.0 license, designating it primarily as a research artifact. While the dataset is freely available, users are responsible for ensuring its legal use in commercial settings. Users must independently verify compliance with applicable laws before employing MINT-1T for commercial purposes.

