# OpenReview forum: "MINT-1T: Scaling Open-Source Multimodal Data by 10x: A Multimodal Dataset with One Trillion Tokens"
_NeurIPS.cc/2024/Datasets_and_Benchmarks_Track — NeurIPS 2024 Track Datasets and Benchmarks Poster_

### Official Review · Reviewer_m5U3 · 2024-07-15
**The paper introduces a large, open-source, interleaved text-image dataset. The dataset has the potential to lead to SOTA open-source multimodal LLMs.**

**Rating:** 8
**Confidence:** 4
**Correctness:** The claims in the submission appear t…
**Clarity:** The paper is well-written and easy to…

**Review:**

The paper presents the largest open-source multimodal interleaved dataset and has the potential to advance SOTA in terms of multimodal LLMs. Assembling a database of this scale (1 trillion text tokens and 3 trillion images) requires significant resources and engineering efforts. The authors have documented all the steps of the process clearly. The paper is of high quality and at the same time easy to read.

In addition to assembling the dataset, the authors perform a thorough analysis of the dataset in terms of diversity, topics covered, topic distribution, etc. Furthermore, they train LLM models to show the utility of the dataset - another significant undertaking.

The results of the models trained from the newly introduced dataset are generally comparable with the results of the model trained from OBELICS.

**Strengths:**

A diverse, large open-source multimodal interleaved dataset is introduced.

The dataset has the potential to improve SOTA for multimodal LLMs.

Significant improvements are shown in terms of multi-image reasoning.

**Additional Feedback:**

It sounds like not all the dataset was used for training LLMs. Is this due to computational resources limitation? It would have been good to see the whole potential of the dataset.

**Documentation:**

The dataset assembly process was well-described. However, a link to the dataset and code does not seem to be available yet, although the authors claims they will make the data (document text and image links) will be made available through Huggingface, if the paper is accepted.

**Ethics:**

The authors construct a very large dataset from HTML, PDF and ArXiv. As for many similar large datasets, the bias in these data sources will be reflected in the MINT-1T dataset.

**Limitations:**

While the authors used some existing strategies to remove inappropriate content, there are no guarantees that all offensive and inappropriate content is removed, as the authors acknowledge.

Some personal identifiable information is removed (e.g., email) but there is still a chance that some is included in the dataset.

The dataset is not self-contained - it only provides links to images and PDFs. While it can be reconstructed, some of the links may not be available at a later time.

**Opportunities For Improvement:**

It would also be interesting to see how the results of the MINT-1T models compare with the results of existing multimodal models, especially in the context of science and technology related tasks.

**Relation To Prior Work:**

Prior work is properly discussed.

**Summary And Contributions:**

The paper aims to assemble the largest open-source multimodal interleaved text-image dataset, called MINT-1T. To achieve this goal, the authors leverage three main sources of data: HTML, PDF and ArXiv and assemble a dataset that contains one trillion text tokens and three billion images. The resulting dataset is 10 times larger than the largest existing dataset, OBELICS. Furthermore, the authors show that using the three sources ensure more diversity than offered by OBELICS, and also good coverage of the Science and Technology domains, as it includes more scientific articles through PDFs and ArXiv. Also notable, special care is taken to remove inappropriate content. To show the utility of the dataset, the authors train LLMs from just the HTLM data (MINT-1T-HTML), all types of data (MINT-1T) and compare results of in-context learning and multi-image reasoning with results of a similar model trained on OBELICS. The results show that the OBELICS, MINT-1T-HTML and MINT-1T models have comparable performance for in-context learning, but the MINT-1T model is superior on the multi-image reasoning task.

---

> ### Author Rebuttal · Authors · 2024-08-17
>
> Thank you for recognizing the significant scale, diversity, and potential impact of our MINT-1T dataset, as well as acknowledging our analysis and engineering efforts.
> > It sounds like not all the dataset was used for training LLMs. Is this due to computational resources limitation? It would have been good to see the whole potential of the dataset.
>
> Yes, running experiments at larger scales was not feasible for us. We are also very excited to see the community use the entire dataset to train very capable multimodal models.
>
> > It would also be interesting to see how the results of the MINT-1T models compare with the results of existing multimodal models, especially in the context of science and technology related tasks.
>
> Thank you for your suggestion; training larger and more capable models on MINT-1T is something we want to do for future work. However, in their current state, our models are base models and have not undergone any instruction tuning. This makes them incomparable to existing models.
>
> > While the authors used some existing strategies to remove inappropriate content, there are no guarantees that all offensive and inappropriate content is removed, as the authors acknowledge.
>
> Since the submission, we have enhanced the content filtering of MINT-1T. In particular, we have improved the safety of MINT by adding very strict image filtering using two NSFW (Not Safe For Work) image classifiers and strict classification thresholds to maximize recall.

---

### Official Review · Reviewer_f6h4 · 2024-07-24
**Review of MINT-1T: Scaling Open-Source Multimodal Data by 10x: A Multimodal Dataset with One Trillion Tokens**

**Rating:** 7
**Confidence:** 3
**Correctness:** The dataset proposed is constructed i…
**Clarity:** The paper is generally well written.

**Review:**

**Quality**: The paper is of high quality with a clear motivation, and the content is relatively complete.

**Clarity**: The paper is logically structured and the analysis is thorough.

**Originality**: The collection and organization of large-scale data are done in a novel way, demonstrating a high level of originality.

**Significance**: The dataset has the potential to become a significant benchmark for future research in the field.

**Pros**:
- The paper addresses the significant engineering challenge of scaling multimodal interleaved datasets, which involves handling larger document sizes and preserving the original ordering of images and text.
- This work is the first in the multimodal interleaved space to collect high-quality multimodal documents on a large scale from diverse sources like PDFs, CommonCrawl, and ArXiv, which enhancing the data diversity.
- The experiments show that LMMs trained on MINT-1T not only match but potentially surpass the performance of models trained on the best existing open-source dataset, OBELICS, while offering a tenfold increase in scale.

**Cons**:
- The document domain distribution in the dataset is not balanced, with varying distributions across different data sources. This imbalance could introduce data biases, potentially affecting the generalization of models trained on this dataset.
- Managing and processing such a massive dataset presents practical challenges, which may limit its accessibility to some researchers.

**Strengths:**

Refer to the Review part.

**Additional Feedback:**

N/A.

**Documentation:**

There is sufficient detail on data collection and organization, availability and maintenance, and ethical and responsible use.

**Ethics:**

N/A.

**Limitations:**

The limitations and potential negative societal impact have been clearly discussed.

**Opportunities For Improvement:**

Plans to release subsets of different sizes will allow researchers to select data appropriate for their specific tasks, facilitating broader accessibility and usability of the dataset.

**Relation To Prior Work:**

The differences between this work and previous contributions are clearly discussed.

**Summary And Contributions:**

This work introduces MINT-1T, a Multimodal INTerleaved dataset comprising one trillion text tokens and three billion images. This dataset addresses the limitations of existing open-source multimodal interleaved datasets, which are smaller, less diverse than their text-only counterparts, and typically sourced only from HTML pages. MINT-1T is the first effort in the multimodal interleaved space to collect high-quality multimodal documents on a large scale from sources such as PDFs, CommonCrawl, and ArXiv. This work also tackles the challenge of managing large-scale data, significantly advancing the development of the field.

---

> ### Author Rebuttal · Authors · 2024-08-17
>
> Thank you for recognizing our work in large-scale data collection, acknowledging the engineering challenges we've addressed, and highlighting the significance of MINT-1T for future research in multimodal training.
> > The document domain distribution in the dataset is not balanced, with varying distributions across different data sources. This imbalance could introduce data biases, potentially affecting the generalization of models trained on this dataset.
>
> We found that using many data sources actually improves the domain balance in MINT-1T. In particular we saw that HTML documents lacked scientific content but had a large representation of other domains. To compensate for this imbalance we add PDF and ArXiv data which contain predominantly science documents.
>
> > Managing and processing such a massive dataset presents practical challenges, which may limit its accessibility to some researchers.
>
> We provide many accessible ways of using MINT-1T. For instance, to enable folks with resource constraints to train on MINT, we released the WebDataset shards for the PDF and ArXiv subsets of [MINT](https://huggingface.co/collections/mlfoundations/mint-1t-6690216ca4d0df7e518dde1c). We have found that models trained on subsets of MINT can also attain strong performance.

---

> > ### Comment · Reviewer_f6h4 · 2024-08-30
> >
> > Thank you for your response. I'll maintain the score.

---

### Official Review · Reviewer_GFBY · 2024-07-24
**A valuable dataset**

**Rating:** 6
**Confidence:** 4
**Correctness:** Yes
**Clarity:** Yes

**Review:**

Strength
1) The ability to process interleaved images and texts is crucial for large multimodal models. It is valuable to provide the community with large-scale multimodal interleaved datasets.
2) A detailed data curation process is presented, although it seems difficult to implement due to the complexity of data engineering. Data analysis shows that MINT-1T has a more uniform distribution than OBELICS.
3) The results on MMMU and Mantis-Eval show that MINT-1T can boost the performance in multi-image reasoning.

Weakness
1)  The dataset is still closed-source until the reviewing process ends, which does not match the rule of Dataset Track.
2) The authors have provided the results of MMMU, which, however, comprises both single-image and multi-image test samples. What about the performance on the multi-image split? Moreover, what about the performance of models pre-trained on MIN1T and OBELICS on recent multi-image evaluation benchmarks such as MileBench [1] and MuriBench [2]?

[1] Dingjie Song et al. MileBench: Benchmarking MLLMs in Long Context.

[2] Fei Wang et al. Muirbench: A comprehensive benchmark for robust multi-image understanding.

**Strengths:**

see review

**Additional Feedback:**

see review.

**Documentation:**

Yes

**Ethics:**

Yes

**Limitations:**

Yes

**Opportunities For Improvement:**

see weakness.

**Relation To Prior Work:**

Yes

**Summary And Contributions:**

This work presents a large-scale multimodal dataset named MINT-1T with interleaved images and texts. Notably, MINT-1T comprises 1T text token and 3B images from diverse sources including HTML, PDF, and Arxiv, which is 10x larger than open-source datasets such as OBELICS. MINT-1T contributes intricate data processing, diverse data sources, and sufficient model experiments. Experiments demonstrate that LMMs trained on MINT-1T not only match but potentially surpass the performance of models trained on the best existing open-source dataset, OBELICS.

---

> ### Author Rebuttal · Authors · 2024-08-17
>
> Thank you for recognizing the value of our large-scale multimodal interleaved dataset, our data curation process, and MINT-1T's potential to improve performance on multi-image reasoning tasks.
> > The dataset is still closed-source until the reviewing process ends, which does not match the rule of Dataset Track.
>
> For this year’s conference, the requirements for the benchmark track is that we release links to the dataset by the camera-ready deadline (see “Submission Instructions” on https://neurips.cc/Conferences/2024/CallForDatasetsBenchmarks). In any case our data is currently open-sourced at https://github.com/mlfoundations/MINT-1T.
>
> > A detailed data curation process is presented, although it seems difficult to implement due to the complexity of data engineering.
>
> As we release the final dataset from our curation process, it is not necessary for users to implement our pipeline and can instead use the data as is. Moreover, we provide WebDataset shards for the PDF and ArXiv subsets of MINT so researchers can train on the dataset without any additional processing.
>
> > The authors have provided the results of MMMU, which, however, comprises both single-image and multi-image test samples. What about the performance on the multi-image split? Moreover, what about the performance of models pre-trained on MIN1T and OBELICS on recent multi-image evaluation benchmarks such as MileBench [1] and MuriBench [2]?
>
> We computed accuracy on multi-image samples from MMMU. In the table attached, MINT-1T refers to the full MINT data mixture while MINT-1T (HTML only) refers to the HTML subset of MINT. We found that the full MINT-1T greatly outperforms OBELICS on multi-image MMMU.
>
> Thanks for your recommendation for other benchmarks we could add! In line with previous works, our primary method for evaluating base multimodal models (that have not undergone any instruction tuning) is through in-context learning benchmarks. Base models are generally not capable of responding to complex instructions that MileBench and MuriBench require without additional instruction tuning. While instruction tuning of base multimodal models is out of scope for this work, we appreciate your suggestion and we want to investigate this in future work.

---

### Official Review · Reviewer_YhHn · 2024-07-25
**MINT-1T Dataset**

**Rating:** 8
**Confidence:** 3
**Correctness:** No correctness issue.
**Clarity:** It is a very well-written paper.

**Review:**

The paper is well-written and easy to follow, with a clear structure that effectively presents the research. This work introduces MINT-1T, the largest open-source multimodal dataset to date. The quality of the research is evident in the meticulous data curation process and the comprehensive experiments that demonstrate the dataset's utility for training advanced LMMs. However there're still some weaknesses on this work, please refer to the strengths and weaknesses for more detailed reviewer's comments.

**Strengths:**

- This paper introduces MINT-1T, an open-source multimodal dataset that is more than ten times larger than existing datasets, featuring one trillion text tokens and three billion images from diverse sources, including previously untapped formats like PDFs and ArXiv papers.
- The paper provides a detailed account of the data curation process, emphasizing quality control through filtering, deduplication, and safety measures to remove sensitive and offensive content, reflecting a commitment to both research integrity and ethical data handling.
- The paper is well-written and easy to follow.
- Experiments support that VLM models trained on MINI-1T exceed the performance of the model trained on the previously released dataset.

**Additional Feedback:**

No additional feedback at this time

**Documentation:**

Tha author claims that the dataset and processing pipeline will be open-sourced and well-documented.

**Ethics:**

No. The dataset underwent several filtering processes as applying NSFW content removal. I didn't see some major concerns here.

**Limitations:**

Yes, the limitations have been discussed.

**Opportunities For Improvement:**

1. The baseline model coverage could have been expanded. For example, if you have enough GPU resources, it would be better to train a model like LLaVA-NeXt-Interleave to further verify the validity of the data you constructed
2. It would be better to provide more quality visualizations of your data
3. The task coverage could have been expanded. You can try to evaluate your data quality through VLM models on more modern benchmarks as MMMU, MathVista, etc.
4. When will this work be open-sourced and the license should be clearly claimed.

**Relation To Prior Work:**

Yes, there is clearly discussed in this paper.

**Summary And Contributions:**

The paper introduces MINT-1T, the largest and most diverse open-source multimodal interleaved dataset to date, featuring one trillion text tokens and three billion images sourced from HTML, PDFs, and ArXiv documents. This tenfold increase in scale over existing datasets aims to bridge the gap in multimodal training data between open-source and proprietary models. The authors detail the data engineering process, highlight the dataset's diversity, and present experiments showing that models trained on MINT-1T can rival or surpass the performance of those trained on the previous leading datasets. The paper also discusses the importance of data safety and the potential for future improvements in data quality and model training.

---

> ### Author Rebuttal · Authors · 2024-08-17
>
> We greatly appreciate your recognition of MINT-1T's unprecedented scale and data curation efforts.
>
> > The baseline model coverage could have been expanded. For example, if you have enough GPU resources, it would be better to train a model like LLaVA-NeXt-Interleave to further verify the validity of the data you constructed.
>
> We were similarly curious about the effects of model architecture on our findings, so we performed ablations with the Idefics2 architecture. Attached are the results on in-context learning benchmarks. We see similar trends to the Xgen-MM model architecture and conclude that our findings hold among different architectures. LLaVA-NeXt-Interleave is another great model but it is slightly different in its training as they focus on instruction tuning and not large-scale multimodal pre-training. Our findings are complementary to LLaVA-NeXt-Interleave as you could use our dataset to pre-train a strong base model and then apply LLaVA-NeXt-Interleave’s instruction tuning on top of that to achieve the best performance (examples of models that do something similar to this are Xgen-MM and Idefics2).
> > It would be better to provide more quality visualizations of your data
>
> This is a great point; thanks for bringing it up! We will add more document visualizations in the appendix for the camera-ready version.
>
> > The task coverage could have been expanded. You can try to evaluate your data quality through VLM models on more modern benchmarks as MMMU, MathVista, etc.
>
> We actually do evaluate on two modern benchmarks, Mantis-eval and MMMU. Results for these are presented in Table 3. One issue with evaluating on modern benchmarks is that they often are geared towards instruction-tuned multimodal models. As our model has only been trained on web-scale documents this is not suitable for instruction based evaluations.
>
> > When will this work be open-sourced and the license should be clearly claimed.
>
> Our dataset is already open-sourced! You can find the data at https://github.com/mlfoundations/MINT-1T. We released the data under a CC-BY 4.0 license. We will add links to the data in our camera-ready version.

---

> > ### Comment · Reviewer_YhHn · 2024-09-09
> > **Official Comment by Reviewer YhHn**
> >
> > Thank you for your response. I'll maintain the score.

---

### Decision · Program_Chairs · 2024-09-26

**Decision:**

Accept (Poster)

**Comment:**

The mulitmodal interleaved dataset (text and image) is deemed to be helpful for MLLM. The authors proposed the largest and most diverse open-source multimodal interleaved dataset MINT-1T, containing one trillion text tokens and three billion images from various sources including HTML, PDFs, and ArXiv documents. And now the dataset has been publicly accessible. Detailed information of the data curation process, emphasizing quality control and ethical considerations, is introduced.  And the performances comparisons with that trained on OBELICS has also demonstrated.

Therefore, I am recommending the acceptance of the paper. And the review comments, such as removing the inappropriate content from the dataset, should be incorporated in the dataset construction.